# Minimal Green Energy Consumption and Workload Management for Data Centers on Smart City Platforms

**Pei Pei [1], Zongjie Huo [1,\*], Oscar Sanjuán Martínez [2] and Rubén González Crespo [2]** 

[1] School of Economics and Management, Lanzhou University of Technology, Lanzhou 730050, China; S3275234@student.rmit.edu.au

[2] Computer Science Department, School of Engineering and Technology, Universidad Internacional de La Rioja, 26006 Logroño, La Rioja, Spain; oscar.sanjuan@ieee.org (O.S.M.); ruben.gonzalez@unir.net (R.G.C.)

\* Correspondence: zongjie@lut.edu.cn; Tel.: +138-9366-8193

**Abstract:** Presently, energy is considered a significant resource that grows scarce with high demand and population in the global market. Therefore, a survey suggested that renewable energy sources are required to avoid scarcity. Hence, in this paper, a smart, sustainable probability distribution hybridized genetic approach (SSPD-HG) has been proposed to decrease energy consumption and minimize the total completion time for a single machine in smart city machine interface platforms. Further, the estimated set of non-dominated alternative using a multi-objective genetic algorithm has been hybridized to address the problem, which is mathematically computed in this research. This paper discusses the need to promote the integration of green energy to reduce energy use costs by balancing regional loads. Further, the timely production of delay-tolerant working loads and the management of thermal storage at data centers has been analyzed in this research. In addition, differences in bandwidth rates between users and data centers are taken into account and analyzed at a lab scale using SSPD-HG for energy-saving costs and managing a balanced workload.

**Keywords:** load balancing; green energy; genetic algorithm; renewable energy

---

## 1. Discussion on Green Energy Consumption and Workload Management

In the present area of research, it is very normal to see that many machines stand idle while a person enters a fabricating plant floor. There can be significant costs to maintain idle machines running in the plant section [1]. In a report by an airline supplier, it was found that, on average, computers stand idle at an 8-h cycle, with 16% of the time. During these inactive times, at least 13% of the total energy consumption is avoided by simply turning the machine off when workers are not handled. In order to save energy, compressors can be used in industrial settings when the machines are inactive, which consumes about 50 percent of the maximum energy. It may be better to shut them off rather than leave them idle for a long time [2]. Turning a machine off when it is not needed and not running a car when waiting for someone can save electricity [3]. The climatic conditions also play an important role in saving electricity, many studies have explored the integration between environmental issues into a decision-making approach [4]. The usage of electricity Can be underestimated when a timetable is set and offers a way of arranging a machine's activities thereby reducing energy use and overall execution period goals by intelligently turning the computer between the scheduled tasks instead of leaving the system idle [5,6]. At present, Internet service providers usually build multiple data centers located in different geographical areas to give information on Web applications, such as social networks [7]. The data centers require significant amounts of electricity to fuel the IT and cooling systems. Data

centers' electricity consumption for web-based applications used 1.5% of the power in 2010, which is expected to rise by about 8% by 2020 [8]. Therefore, Internet service companies have been making intensive efforts to reduce their data centers' electricity costs. [9,10] Moreover, the desire of Internet service providers is rising to be "sustainable," allowing them to reduce their environmental impact to the economic impact of their data centers. The worldwide electricity generated from fossil-fuel plants, such as coal and gas power stations, provides two-thirds of electricity generated by the electricity supply system [11]. Hence, the renewable generators, being popular in construction costs, become more and more desirable choices for running data centers, particularly as policy measures promote renewable energy [12]. By comparison to current brown electricity from energy networks, though, green energy coming from renewable sources, such as wind and sun, is unreliable and uncontrollable, creating a major challenge for data centers to use is efficiently.

The main challenge is the task of managing electricity supply and demand instantaneously, which can particularly resolve the issue of green energy with large-scale electric energy collected from renewable sources, as shown in Figure 1, based on the generation, transmission and distribution systems. Although it is helpful to manage geographical loads, there are two potential possibilities for promoting the introduction of renewable energies in data centers [13], which is coordinated based on the control center. In the control center, the difference between an interactive workload and batch workload follows two aspects [14]. The first aspect are computational requirements for the interactive workload which is small and batch workload is large [15]. Further, the second aspect is the response time that refers to the output system for the virtual workload [16]. Hence, it is the overall workload within a period of time. The important point for the cooling system offers a large share of electricity consumption in a data center [17]. If the price of the power is high, the stored resources can be used to support the data center to cool down for energy saving [18]. The examination is done with the issue of shared regional charge, balance time tolerance preparation, and workload management in geographically distributed data centers. They often take the connectivity expense for cloud customers and data centers into consideration, as well as the brown energy costs. The major aim of this research is to reduce the energy emissions of data centers through sustainable energy management at data centers with less delay tolerance. Renewable energy is typically sporadic and volatile, in comparison to conventional electricity energy. Whether to best utilize green electricity in data centers from such clean sources is a problem. In this paper, to promote renewable energy adoption and minimize costs of energy usage by spatial load balancing, opportunistic scheduling of delay-tolerant workloads control in data centers has been accomplished using SSPD-HG. This approach focuses on a multi-objective framework for energy consumption and to minimize the total completion time for a single machine. Furthermore, in this paper, bandwidth rates between users and data centers are analyzed using SSPD-HG for energy-saving and managing a balanced workload. The rest of the paper is organized as follows: Section 2 reviews the related work, Section 3 provides insights regarding our proposed SSPD-HG approach, further, in Section 4 mathematical model for green energy consumption and workload management for data centers is analyzed, in Section 5, the findings of the study related to SSPD-HG work are discussed. Finally, the paper closes with its conclusion and future scope.

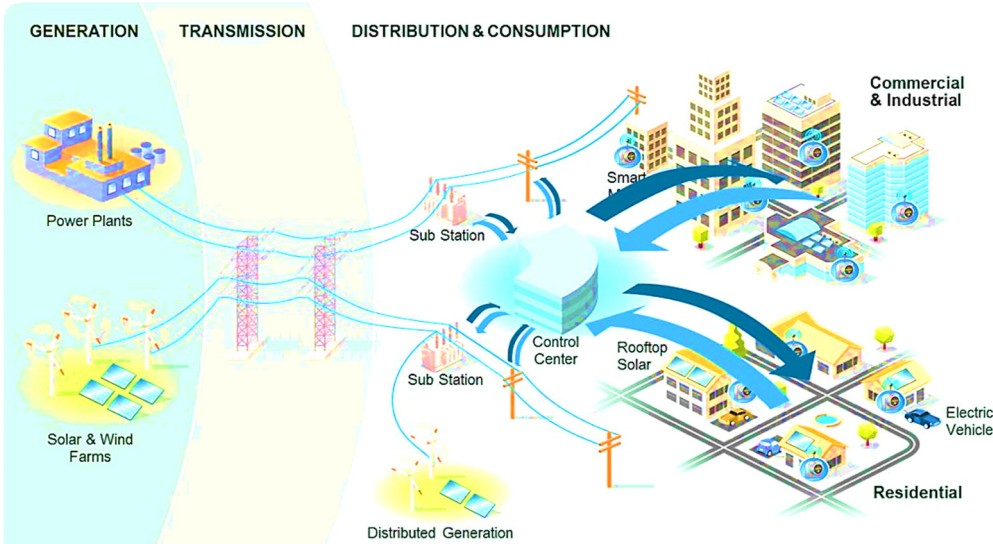

**Figure 1.** Large-scale electric energy collected from renewable energy sources.

## 2. Background Study on Workload Management in Data Centers and Minimal Energy Consumption

Many research works have been carried out [19,20] where the authors discussed the mathematical model to minimize the energy consumption and decrease the overall completion time of one machine. Further authors reported a multi-objective genetic algorithm to achieve minimal energy consumption at the data centers. The mathematical model can change significantly on a computer when other design priorities are mathematically analyzed, hence, based on the solution strategy the exception of the linear system that produces the estimated front of Pareto can adjust the genetic algorithm according to the sort of programming goal. In [21] the author proposed the use of the opportunities offered by global charge equilibrium, the ability to such plan workloads for delay tolerance and the control of thermal storage in data centers, can promote the improvement of renewable energy. In fact, differences in bandwidth rates are recognized between customers and data centers [22]. This paper illustrates how to choose the lowest range of active base stations (BS's), which can sustain the service quality (the minimum data speed) desired by consumers, "as a consequence of energy consumption minimization (ECM)". The EMC algorithm empties the collection of active BSs and attaches one by one based on BS accordingly [23,24]. Cloud data centers are resources that consume significant amounts of energy and the migration technology of the virtual machine (VM), by consolidating virtual machines on a minimal number of servers, which can be applied to reduce energy consumption [25]. A new model of energy efficiency has been set up that formulates and integrates device costs [26], transfer costs, and relocation costs [27,28], which has been reported by several researchers. Two additional heuristics for VM placement have been introduced and the examination of the time complexity indicates that the algorithms presented are scalable for optimizing delay tolerance and the management of thermal storage at data centers. The implementation of the proposed algorithms in a specific cloud platform can be taken into account for energy consumption, VM migration drawbacks may include enhancing the overuse prediction algorithms for lower social level agreement (SLA) violations induced by overloaded servers [29]. Hence, the high renewable energy penetration, for the development of energy-efficient cooling systems and flexibilities can be widely used for the energy storage system. Through the use of stochastic modeling techniques, it is suggested that a centralized management approach allows data centers, in compliance with long-term service quality of service (QoS) criteria, to respond to irregular renewables demand, coolant volatility, IT workload shifts, and energy prices [30]. Furthermore, potential approaches will include modeling techniques with more realistic electricity leakage storage units, including the cost of energy transport from storage units, as well as the layout of the power

network transmission. To overcome all these drawbacks, SSPD-HG approach mainly focuses on minimal energy consumption and reducing the total completion time for a single machine. Furthermore, in this paper, bandwidth rates between users and data centers are analyzed for energy-saving and managing a balanced workload using the SSPD-HG approach.

## 3. Minimal Energy Consumption and Total Completion Time Mathematical Model (SSPD-HG)

The representation of task or job is given by $l$. m and n have been modeled for each job, where the machinery specific data, such as total power ($strength_{total}$) and install energy ($strength_{install}$), ($strength_{rest}$), have been used in the machine. The total completion time is modeled and can be represented in the mathematical Equation (1). Hence the ($strength_{total}$), as shown in Equation (3), is the total power consumed by the machine during on condition, ($strength_{install}$), is defined as the turn off and then turn on (sequence) condition of the machine. ($strength_{rest}$) is the idle state condition for the machine, which has been processed based on the processing state.

$$min\left(\sum_{m=1}^{x} q_m\right)strength_{total} + \left(_m^{max}d_m - _m^{mim}d_m\right) * strength_{rest} - \sum_{m=1}^{x}\sum_{n=1 \neq m}^{x} y_m n + strength_{install} \tag{1}$$

$$strength_{total} = thetotalpowerconsumedbymachine \tag{2}$$

$$strength_{total} = strength_{working} - strength_{idle} \tag{3}$$

$strength_{working}$ is the power consumed by a machine during working conditions.

- $(max_m d_m - min_m d_m * strength_{rest})$ is the total rest power of sequence of jobs.

- $min\left(\sum_{m=1}^{x} q_m\right)$ is the main objective of total completion and total energy consumption time.

- $y_m n$ is the installation and the total marginal energy consumption between jobs m and n.

The goal of SSPD-HG is listed as follows:

(i) To demonstrate the effects of thermal storage together with accommodating workload management in data centers, to reduce working costs, and

(ii) To get clear clarification between cost savings, working load delays and thermal storage capacities and the information has been taken from Google data centers. The output distribution is relative to the distances between cities and energy costs between proxies and data centers based on the buffer and non-buffer, condition as shown in Figure 2. $strength_{idle}$ is the energy consumption for server in one period and it is given by $strength_{idle}$= 100 W *1/6 h and $strength_{working}$ is the busy state and the energy consumption of each server $strength_{working} = 250W * 1/6$ h.

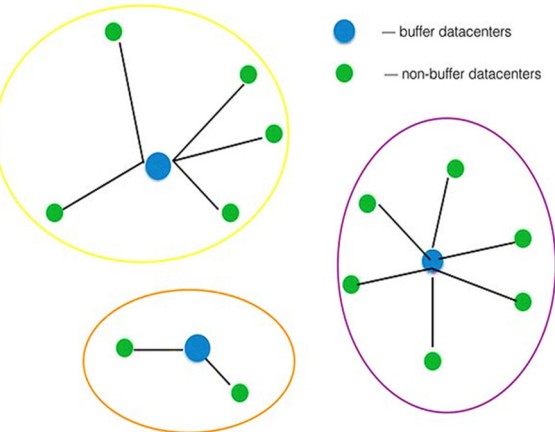

**Figure 2.** Large-scale electric energy collected from renewable energy.

SSPD-HG approach mainly deals with the total energy consumption and workload management of data centers. Hence, by considering a cloud service company with workload management and data collection, which are shown in Figure 3 submission for job or operation first hits the proxy k and the proxy will determine the data center to handle the job application. The proxy does not have a storage buffer based on a request which enters the proxy, where a data center can be diverted for immediate retrieval.

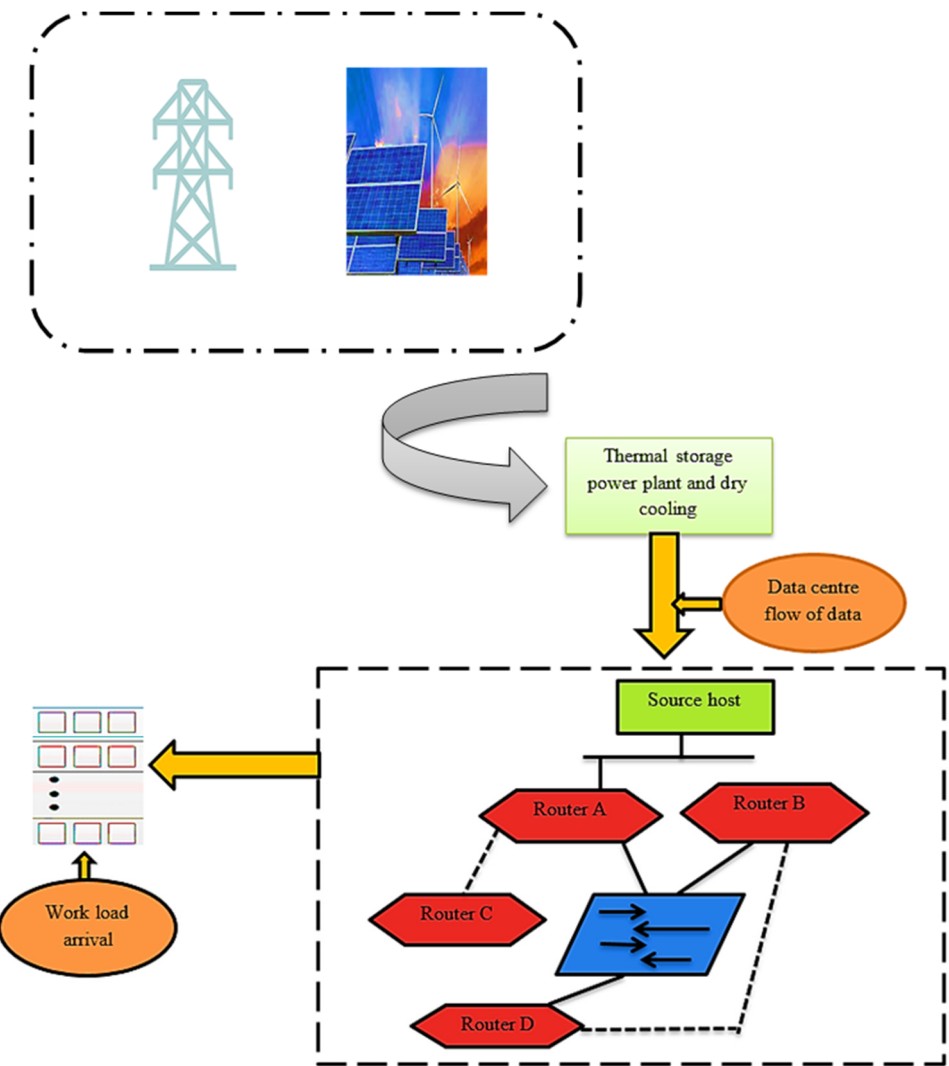

**Figure 3.** Data collection of SSPD-HG and workload management.

The traces of wind and solar energies are gathered for every 10 min and the speed of the wind and solar radiation are measured based on the workload management. The traces are well-scaled hence the total renewable energy generation in each data center will reach half the average energy consumption. The first two days show a share of solar and wind energy in two regions and the mathematical model can be represented in Equation (4)

$$\sum_{k=1}^{K} X_k^a(t) \leq X_{max}^a \forall_a, t \tag{4}$$

At the end of proxy $k$ the work or job request arrives. Here the $X_k^a(t)$ is the time for proxy $k$ with the type of job, where the rate of arrival of the workload is denoted as $X(t) = X_{max}^a$, $\forall_a$, $t$. Hence, the total arrival work of job $a$ is given by constant value $X_{max}^a$. Here, each data center has a thermal storage system installed based on the optimal section. During this time, the optimal charge and discharge

conditions have the system that becomes the peak refreshing energy consumption, which has been shown in Figure 4.

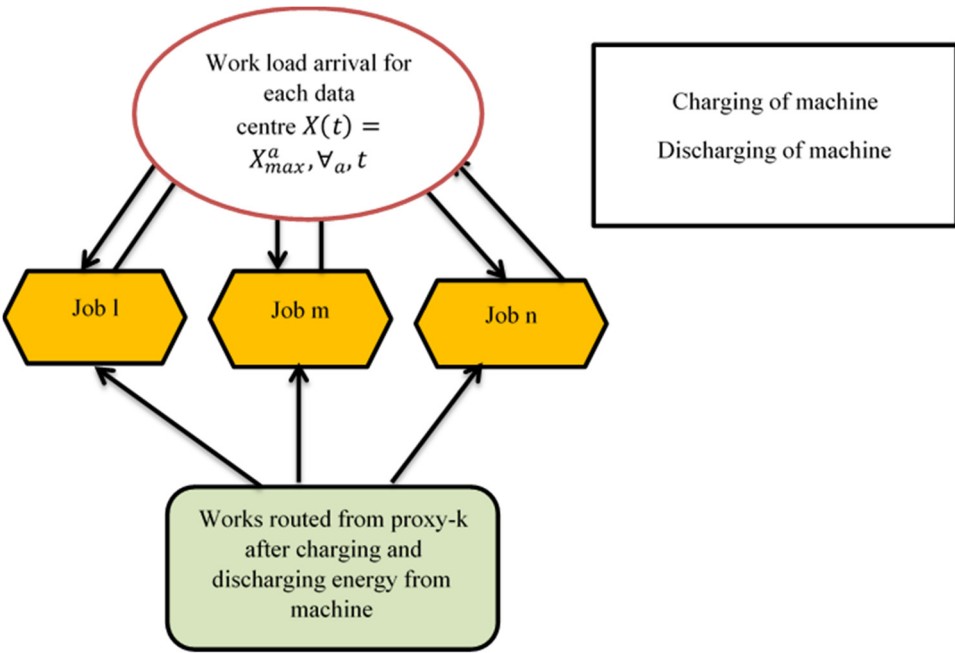

**Figure 4.** Charging and discharging of machine for each proxy of work.

The number of the type of work which is denoted by *a* reaching at proxy *k* in time *t* as $X_t^a$. The work arrival rate vector at time t is denoted as $X(t) = \left( x_t^a(t), \forall_a, t \right)$ and the time average of the vector is given by $= E\{X(t)\}$. Hence, the finite and the optimistic constant to $X_{max}^a$ needs to be limited for the cumulative arrival rate of form *a* workers.

$$\sum_{l=1}^{M} \lambda_{lk}^a(t) = X_k^a(t), \forall_a, k, t \tag{5}$$

$$\lambda_{lk}^a(t) \geq 0, \ \forall_a, l, k, t \tag{6}$$

$\lambda_{lk}^a(t)$ is used to represent the number of works that are routed from proxy *k* to the data center *l*, in time *t*, $\lambda_k^a(t) = \lambda_{lk}^a(t)$, where $\forall_l$ represents the routing vector for the *a* type of work at proxy k, which has been modeled in Equations (5) and (6). Notice that previous studies focus primarily on reducing energy costs by taking bandwidth costs into account for routing workloads.

In addition to the cost of using thermal storage devices, as described earlier, there are two main aspects for the total operating costs which are listed as follows,

- One is the cost for the energy consumption in data centers and
- The other is the cost of connectivity between consumers in the proxies and data centers in the cloud, as represented in Equation (7).

$$min \sum_{m=1}^{x} d_m. \tag{7}$$

$$d_m - q_m \geq z_i \forall_m = 1 \ldots .x \tag{8}$$

Based on the mathematical formulation and the statistic that of SSPD-HG beats all benchmark schemes, as shown in Equation (8). In fact, the thermal storage can help to reduce the total

electricity cost, while SSPD-HG takes into consideration the time-limited price of electricity, this is intermittent and does not work until the energy price is low enough. The algorithm for the SSPD-HG algorithm is shown below and mathematically formulated based on the total completion time.

---

**Algorithm 1.** SSPD-HG algorithm.

---

*Input:* $\boldsymbol{d_n, q_m, d_m, T_D}$
*Output:* $\boldsymbol{y_{mn}}, \boldsymbol{\forall_m, \forall_n}$
If $((\mathbf{d_n - q_m}) - \mathbf{d_m}) - \mathbf{T_D}$
$\mathbf{y_m n} = ((\mathbf{d_n - q_n}) - \mathbf{d_m}) * \mathbf{strength_{rest}}$
Else $\mathbf{y_{mn} = 0}, \forall_m = \mathbf{1} \ldots \mathbf{x}, \forall_n = \mathbf{1} \ldots \mathbf{x} \neq \mathbf{m}$
*End if*
$d_n - q_n \geq d_m$ *or* $q_n \leq d_m - q_m,$
*Case1: if*
$\forall_m = \mathbf{1} \ldots \mathbf{x}$
$\forall_n = \mathbf{1} \ldots \mathbf{x} \neq \mathbf{m}$
$\mathbf{d_j, y_{mn} \geq 0}$
*Else*
$\forall_m = \mathbf{0}$
$\forall_n = \mathbf{0}$
$\mathbf{d_j, y_{mn} \geq 0}$
*Close*

---

As inferred from Algorithm 1, the main objective of the total completion time of the process is $\sum_{m=1}^{x} d_m,$ and the complete strength consumption can be given as $(\sum_{m=1}^{x} q_m) strength_{total} + (max_j c_j - min_j c_j) *$ $strength_{rest} - \sum_{m=1}^{x} \sum_{n=1 \neq j}^{x} y_{jk} + strength_{install}.$ Hence, the strength install $((strength_{install} = strength_{totla} \sum_{j=1}^{n} p_j)$ is constant and the total amount of strength of the machine at rest condition between the completion of first and last jobs is $(max_m d_m - min_m d_m) * strength_{rest}.$ Instead, when taking bandwidth costs into account, the SSPD-HG algorithm with the basic systems has the cost for bandwidth, which is negligible in our situation because all operating loads are diverted to the closest data centers. SSPD-HG can achieve the largest total operating cost savings by considering the different bandwidth costs between proxies and data centers.

The SSPD-HG approach combines two targets into one target, which can be achieved by applying the weighted average of both goals. The mathematical model for the weighted problem's objective function is given in Equations (9) and (10)

$$h(\varnothing_1, \varnothing_2) = \varnothing_1 h_1 + \varnothing_2 h_2 \tag{9}$$

$$h(\varnothing_1, \varnothing_2) = \varnothing_1 \sum_{k=1}^{x} d_m + \varnothing_2((\sum_y ((d_m + 1 - q_m + 1) - d_m * \\ strength_{rest} + \omega * energy_{install} \tag{10}$$

In this segment, the compensation is the main focus within SSPD-HG for time, total operating costs and thermal storage power. The associated total operating cost and estimated workload delay are determined by the various choices. However, the SSPD-HG will reduce its total operating costs with compensations for the working load period in which the study results are checked in the theorem by increasing the parameter.

Apart from the expense of using a thermal storage network as previously described, two other aspects are compensated by the total cost: One is the cost of energy used for functioning in data centers, and the other is the cost of connectivity for consumers close to cloud proxies and data centers. They

believe that the marginal cost of producing renewable energy is zero, thereby encouraging the data centers to use as much as feasible, to encourage the use of green energy from renewables.

The quality of conventional brown energy generated from the grid ranges from place to time and relies on the wholesale power sector. $q_i(t)$ denotes the renewable energy price that is purchased in time $T$ at the $DC_i$ data center on the wholesale electricity market varies in time and depending on the venue.

Hence, it is analyzed that $0 \leq q_i(t) \leq q_i^{max}$ for all periods $T$ and $q_i^{max} \geq \gamma_i / \eta_i$ is sectored based on the total power consumption of the data center, which can be related to Equation (11)

$$(1- \propto)p_i^{rest} + \propto p_i^{working} \tag{11}$$

where $p_i^{rest}$ power consumption in the idle state of the system, $p_i^{working}$ power consumption in the working state of the system.

## 4. SSPD-HG Approach

The algorithm for SSPD-HG is given as follows,

1.  The thermal energy queues analyze this continuously based on the unpredictable circumstances as denoted as $(s_l(t), q_l(t)X_k^a(t), \forall_a, l, k)$, which can be modeled in Equation (12)

$$0 \leq R_i(t) \leq R_i^{max}, \forall_l \tag{12}$$

2.  The vector $(s_l(t), q_l(t)X_k^a(t), \forall_a, l, k)$ is the periods and constant $\varnothing$ such that $\delta + \omega 1 \epsilon \Omega$ has the workload line length that satisfies the unpredictable circumstances process.

$$\overline{M} \leq \frac{B_1 V + B_2}{\omega} \tag{13}$$

$$0 \leq R_i(t) \leq R_i^{max}, \forall_l \tag{14}$$

The test SSPD-HG under specific $(s_l(t), q_l(t)X_k^a(t), \forall_a, l, k)$ for assessing thermal storage costs in terms of operating cost efficiency. The energy queue for each job and the processing state are shown in Figure 5. The operational cost saving is lower with the rise of the thermal storage cost factor. SSPD-HG does not use thermal storage at all, and the delay resistance scheduling based on the regional load balancing for the contribution of costs savings

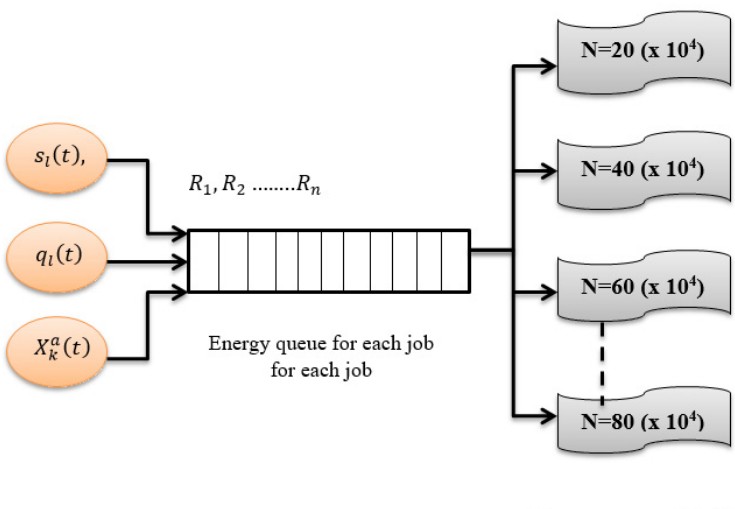

**Figure 5.** Energy queue for each job and the processing state.

Developing an SSPD-HG approach for a certain chromosome would significantly improve the speed of the algorithm. Once the sequence of the work is set, it reduces the total completion time as soon as it is available based on the idle state conditions of the machine. For this particular solution, the SSPD-HG method has an optimal completion time and high energy usage. Certain non-dominated solutions are available by slowly decreasing energy consumption rates (despite the overall completion period increase) and by reducing the total completion time from the solution and moving the jobs back after the implementation of their completion time to the starting time of the next mission. This method decreases the idle time-consuming resources and raises the start-up time based on the configuration energy, which is independent of the length of the setup cycle.

The development of heuristic solutions for a certain chromosome would increase the performance of the algorithm substantially. When the order of the task is set, the next job is completed if necessary. Further, the machine stays idle, hence, it further decreases the overall completion time because the order of work is known. This solution has an optimal completion time for this particular solution and comparable energy consumption based on the idle time-consuming resources. Many non-dominated solutions can be achieved by reducing energy consumption gradually (although the total completion time can increase), beginning with the solution that minimizes the overall completion time and moving the jobs back after deployment until the start time of the next job corresponds to their completion times. After implementation, all jobs go back to a third job, and continue to another set-up. In addition, differences in bandwidth rates between users and data centers are taken into account and analyzed at lab scale using SSPD-HG for energy-saving costs and managing a balanced workload, which is discussed below.

## 5. Results and Discussion

Through increasing the crossover rate from 0 to 2, the output metric varies according to the crossover rate. The optimum convergence rates vary according to the number of jobs listed, the skill for a low, medium, and high number of jobs with the same characteristics (20, 40, and 60, respectively), as shown in Table 1.

**Table 1.** Total number of crossover and total number of jobs.

| Rate of Cross over | N = 20 ($\times$ 104) | N = 40 ($\times$ 104) | N = 60 ($\times$ 104) |
|:---:|:---:|:---:|:---:|
| 0 | 6.542 | 2.632 | 5.213 |
| 0.2 | 7.856 | 1.925 | 4.653 |
| 0.4 | 6.565 | 1.825 | 3.256 |
| 0.6 | 7.120 | 1.725 | 4.988 |
| 0.8 | 6.253 | 1.956 | 4.215 |
| 1.0 | 6.4654 | 1.985 | 4.325 |
| 1.2 | 5.6456 | 1.8456 | 3.564 |
| 1.4 | 5.647 | 1.564 | 3.562 |
| 1.6 | 6.988 | 1.243 | 3.6256 |
| 1.8 | 5.999 | 1.645 | 4.265 |
| 2.0 | 6.898 | 1.856 | 4.653 |

The optimum crossover rate of 0.8 numerical factor has been analyzed with the same values that allow us to perform the same type of experiment to determine the optimum number of individual's K in a generation as a function of the number of works to be done, as shown in Figure 6. Hence, a smaller batch, 25 with reasonable solutions of consistency, has been chosen and depicted in Figure 6.

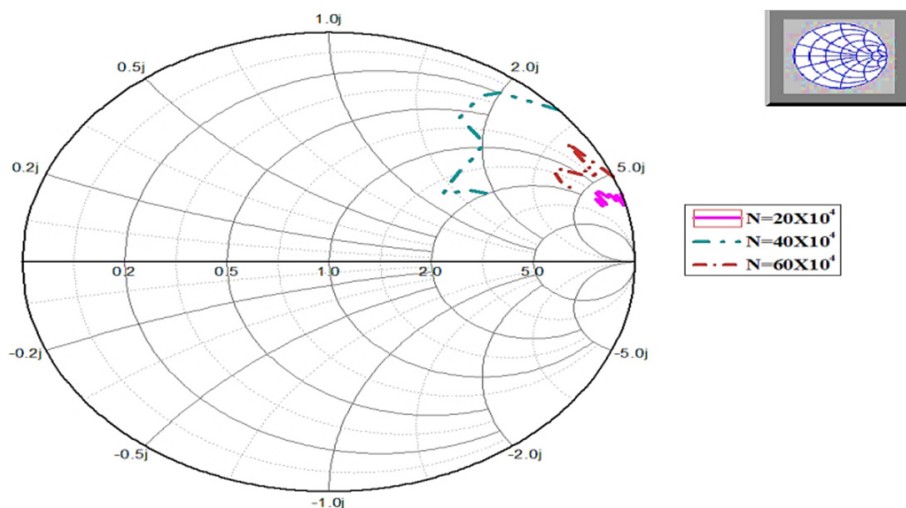

**Figure 6.** Analysis of N = 20 × 104, N = 40 × 104, N = 60 × 104.

If there is a change in the number of generations, the rate of convergence and the number of people within a generation, it can be seen that the optimal number of generations exceeds the highest allowable number of people, as shown in Figure 7. It ensures that the number of generations is reduced to 40 years.

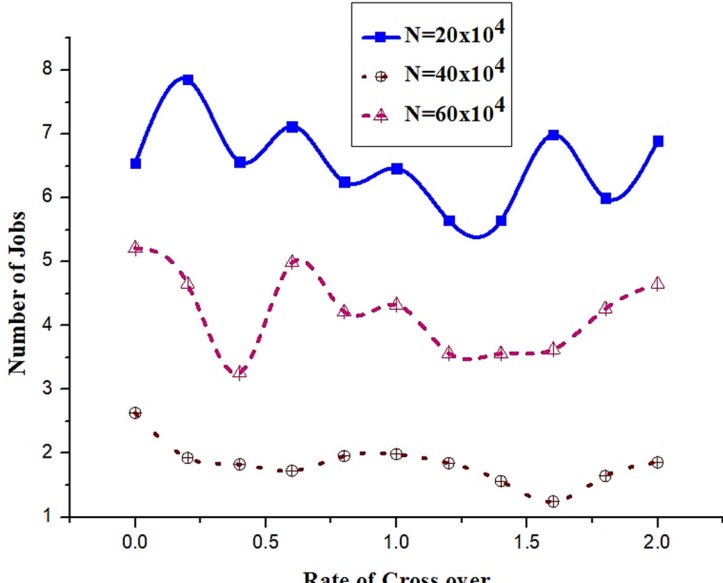

**Figure 7.** Rate of crossovers vs. number of jobs.

In the change in the number of generations and the number of people in a group, we can see that the optimal number of generations is at the maximum allowable amount of 200 as shown in Table 2. Since the thermal storage can help to reduce the total electricity cost, while SSPD-HG takes into consideration the time-limited price of electricity, this is intermittent and does not work until the energy price is low enough.

**Table 2.** Number of generation and the total number of jobs.

| K | N = 20 (× 104) | N = 40 (× 104) | N = 60 (× 104) |
|---|---|---|---|
| 20 | 7.8562 | 2.3526 | 4.652 |
| 40 | 6.524 | 1.675 | 4.562 |
| 60 | 5.952 | 1.865 | 3.1526 |
| 80 | 6.532 | 1.956 | 3.256 |
| 100 | 6.452 | 1.56 | 3.4235 |
| 120 | 8.563 | 1.456 | 3.325 |
| 140 | 7.652 | 1.788 | 3.352 |
| 160 | 8.152 | 1.88 | 3.2567 |
| 180 | 7.652 | 1.7765 | 3.967 |
| 200 | 8.652 | 1.7568 | 4.0125 |

As a result, the number of generations will be limited to 60 generations, as shown in Figure 8. SSPD-HG runs using the second fitness function with specific number of jobs are evaluating the optimum value of the other parameters. With the optimal set of parameters, we can use the same parameters to equate two fitness functions and adjust the number of jobs.

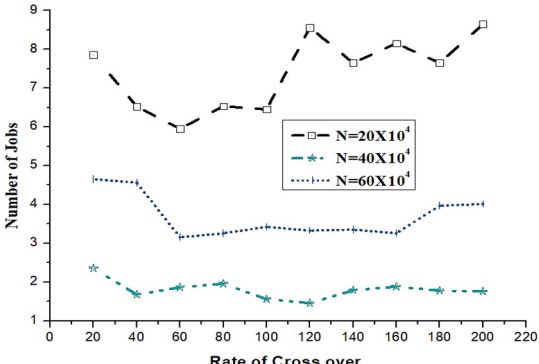

**Figure 8.** Rate of crossover vs. number of jobs.

SSPD-HG is the approach that permits a consumer to solve the total completion time of a job or task by the machine and the workload management. Other issues with overlapping goals in different operating conditions (i.e., with specific preparation priorities and multi machines) need to be addressed. The total number of jobs and the rate of crossover gives the workload management of the data centers for each job, as shown in Figure 9

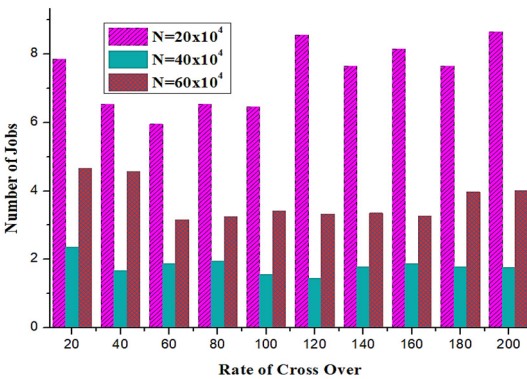

**Figure 9.** Rate of cross over vs. number of jobs.

Total completion and total energy target varied between 0 and 4260, respectively (where all jobs are scheduled for the maximum start date after work). The ratio has bene chosen between a collection of non-dominated options based on different parameters and sub-criteria, for the best alternative solutions. Hence, the number of generations for each job is shown in Table 3.

**Table 3.** Total Number of generation Vs Number of Jobs.

| Generation Number | N = 20 (× 104) | N = 40 (× 104) | N = 60 (× 104) |
|---|---|---|---|
| 20 | 9.542 | 8.956 | 5.264 |
| 40 | 8.568 | 7.652 | 5.265 |
| 60 | 6.253 | 6.526 | 4.256 |
| 80 | 5.265 | 6.586 | 4.2366 |
| 100 | 5.3265 | 6.325 | 3.256 |
| 120 | 5.1425 | 6.458 | 3.256 |
| 140 | 5.125 | 5.565 | 3.564 |
| 160 | 4.568 | 5.242 | 3.458 |
| 180 | 4.652 | 5.155 | 2.965 |
| 200 | 4.864 | 5.235 | 2.846 |

The decision-maker requires a hierarchy of requirements and subscriptions to organize the problem. In order to determine the importance of each option for consideration, the subject must also provide a summary of the parameters to pairs. The comparison is done between the number of jobs and number of generations. Instead, when taking bandwidth costs into account, the SSPD-HG algorithm with the basic systems has the cost for bandwidth, which is negligible in our situation because all operating loads are diverted to the closest data centers. Hence, the SSPD-HG can achieve the largest total operating cost savings by considering the different bandwidth costs between proxies and data centers.

The production management is able to select from the 105 non-dominated solutions based on their choice, using the 220-employee question, as shown in Figure 10. Here, each data center has a thermal storage system installed based on the optimal section based on the number of generation vs. jobs. During this time, the optimal charge and discharge condition has the system that becomes the peak refreshing energy consumption.

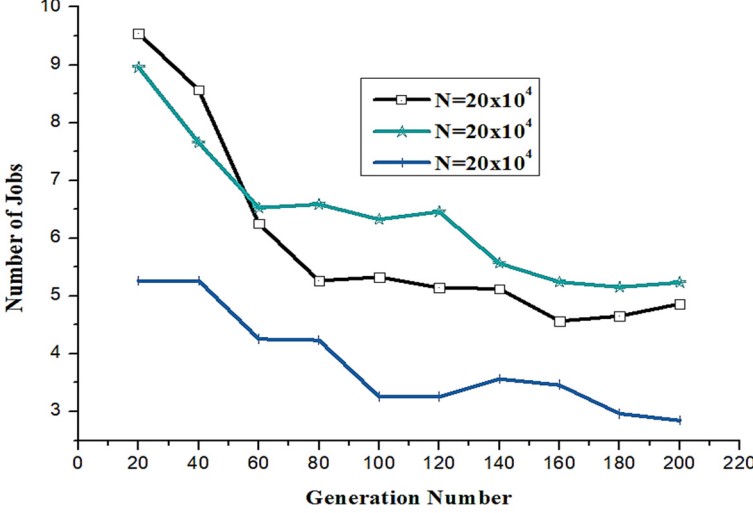

**Figure 10.** Number of generation vs. number of jobs.

## 6. Conclusion and Future Work

This study provides a guide for manufacturing practitioners to incorporate green engineering planning and sustainable technologies in an age where environmentally-friendly industrial practices are very valued. Our purpose is to build a mechanism through which a decision-maker could choose the most effective schedule at a reasonable level of energy consumption. The problem of a micro combination of total energy consumption and total production time is a complicated one, and it takes considerable time to find the right solution. The developed multi-target genetic algorithm provides a good approximate value in a reasonable time. Hence, the SSPD-HG is the first approach to solve the total energy completion time problem for a device. Nevertheless, other issues with overlapping priorities in different operating environments (i.e., different programming goals and several machines) need to be explored. For instance, where consideration is given to other scheduling objects, where formulations can be dramatically changed on a single machine. Hence, the approach to the solution will be identical with the exception of an estimated Pareto front scheduling system and the SSPD-HG approach may vary according to the type of plan purpose.

**Author Contributions:** Conceptualization, Z.H.; Formal analysis, P.P. and R.G.C.; Investigation, P.P. and Z.H.; Methodology, R.G.C.; Project administration, Z.H.; Resources, P.P., Z.H. and O.S.M.; Software, Z.H.; Supervision, O.S.M.; Validation, R.G.C.; Visualization, P.P.; Writing—original draft, Z.H. All authors have read and agreed to the published version of the manuscript.

**Funding:** This research received no external funding.

**Conflicts of Interest:** The authors declare no conflict of interest.

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
