# Peer review of "Minimal Green Energy Consumption and Workload Management for Data Centers on Smart City Platforms"

_sustainability, doi:10.3390/su12083140_

Round 1

Reviewer 1 Report

This manuscript does an excellent job in demonstrating significant estimated set of
non- dominated alternative using a multi-objective genetic algorithm based on
hybridized problem which is mathematically computed in this research. Since each data
centre has a thermal storage system installed based on the optimal section. During this
time, the optimal charging and discharge condition has not been mentioned based on
the peak refreshing energy consumption as shown in figure 3.    Furthermore, Notice that
previous studies focus primarily on reducing energy costs by taking bandwidth costs
into account for routing workloads. In addition to the cost of using thermal storage
devices, as described earlier, there are two main aspects for the total operating costs
which is not derived based on mathematical formulation. It should now be possible to
perform selection experiments from the original producing renewable energy to replicate
the founding event and estimate the heritability of these traits.     The manuscript does not
contribute towards further alternatives to easily visualize high dimensionality data on the
web. It’s simple and easy to embed into other web frameworks or applications where
authors should pay their concern. Instead, when taking bandwidth costs into account,
The SSPD-HG algorithm with the basic systems has the cost for bandwidth is more
negligible and in what way it is more opera-table based on the closest data centres.
SSPD-HG can achieve the largest total operating cost savings by considering the
different bandwidth costs between proxies and data centres, whereas the bandwidth
range has not been mentioned in the results and discussion.

Author Response

A submission for job or operation first hits the proxy k and The proxy will determine the data centre to handle the job application. The proxy does not have a storage buffer based on a request which enters the proxy, where a data centre can be diverted for immediate retrieval.

.This study provides a guide for manufacturing practitioners to incorporate green engineering planning and sustainable technologies in an age where environmentally-friendly industrial practices are very valued. Our purpose is to build a mechanism through which a decision-maker could choose the most effective schedule at a reasonable level of energy consumption. The problem of a micro combination of total energy consumption and total production time is a complicated one, and it takes considerable time to find the right solution. The developed multi-target genetic algorithm provides a good approximate value in a reasonable time.

Reviewer 2 Report

A brief summary

The article starts with stating that many machines stand idle at a fabricating plant flour. To reduce energy and energy costs, they can better be switched off when they are not used. Then the authors turn to data centres, used by internet providers, which use a huge amount of energy. Probably the amount of energy can not be reduced at the moment, but the electricity costs and the energy source can. Concerning the latter, from a point of view of sustainability it is good to turn from brown energy (based on fossilfuel plants) to green energy (renewable energy, like wind, water and solar energy). But green energy is unreliable and uncontrollable. Though the authors do not elaborate why green energy is ‘unreliable and uncontrollable’, they come up with a solution for data centers: minimize energy consumption and to decrease the total completion time for a single machine. This is elaborated in the rest of the article in a mathematical way by introducing a smart sustainable probability distribution hybridized genetic approach (SSPD-HG).

According to me, it is an interesting topic, but it is written in a very technical manner.

  • Broad comments

Paragraph 1, which is the introduction to the article, should be elaborated more,  concerning definitions of important constructs. Furthermore, not all ‘thoughtsteps’ taken by the authors  are clear in the text (see also my specific comments).

What I miss are definitions of important constructs in the article, like ‘sustainability’, ‘interactive workload’ and ‘batch workload’.

Please look after the English. There are sentences in the text which I do not understand.

  • Specific comments

Line 11: The authors state ‘ Presently, Energy is considering a significant resource that grows scarce with high demand and population in the global market. Therefore, a survey suggested that Renewable energy sources are typically unreliable in accordance with traditional energy sources.’ The second sentence does not logically follow on the first sentence (Why ‘therefore’?).

Line 47: The construct ‘sustainable’ hase to be defined and elaborated more.

Line 52: The authors write ‘By comparison to current brown electricity from energy networks, though, green energy is unreliable and uncontrollable from renewable sources, (…)’. Please elaborate why green energy is unreliable and uncontrollable. This is very important, as the authors then come up with a solution fort his problem. But the problem is not clear to me yet.

Line 61: The author write ‘Although it is helpful to manage geographical loads, there are two potential possibilities for promoting the introduction of renewable energies in data centres which is coordinated based on the control center. The difference between an interactive workload and batch workload follows two aspects’. Here the authors go too fast. How is the laste sentence related to the former sentence?

Line 63: Please define what is meant with ‘interactive workload’ and ‘batch workload’.

Line 68: What is meant with ‘stored resources’? You mean energy storage?

Line 95: With ‘IMC algorithm’is meant ‘EMC algorithm’? Otherwise, the authors have to write first in full what is meant by ‘IMC algorithm’.

Line 104: Please write in full first the abbreviation SLA.

Line 118: The authors write ‘The representation of task or job is given by?, m, and n has been modelled’. What does l, m and n stand for? And how do the authors come up with these ‘dimensions’? Where are they for example derived from?

Line 127: Please elaborate more what is meant with ‘the total rest power of sequence of jobs’.

Line 136: The authors write ‘(…) the information has been taken from Google data centres’. Please explain from the point of view of methodology how the authors did take this information.

Line 244: What is, in this context, meant with a ‘chromosome’?

Author Response

The article starts with stating that many machines stand idle at a fabricating plant flour. To reduce energy and energy costs, they can better be switched off when they are not used. Then the authors turn to data centres, used by internet providers, which use a huge amount of energy. Probably the amount of energy can not be reduced at the moment, but the electricity costs and the energy source can. Concerning the latter, from a point of view of sustainability it is good to turn from brown energy (based on fossilfuel plants) to green energy (renewable energy, like wind, water and solar energy). But green energy is unreliable and uncontrollable. Though the authors do not elaborate why green energy is ‘unreliable and uncontrollable’, they come up with a solution for data centers: minimize energy consumption and to decrease the total completion time for a single machine. This is elaborated in the rest of the article in a mathematical way by introducing a smart sustainable probability distribution hybridized genetic approach (SSPD-HG).

According to me, it is an interesting topic, but it is written in a very technical manner.

Author’s response: The High renewable energy penetration has been elaborated, for the development of energy-efficient cooling systems and flexibilities can be widely used for the energy storage system. Through the use of stochastic modelling techniques, it is suggested that a centralized management approach allows data centres, in compliance with long-term service quality of service (QoS) criteria, which respond to irregular renewables demand, coolant volatility, IT (IT) workload shifts and energy prices. Furthermore, Potential approaches will include modelling techniques with more realistic electricity leakage storage units, including the cost of energy transport from storage units, as well as the layout of the power network transmission. To overcome all these drawbacks, SSPD-HG approach mainly focuses on minimal energy consumption and to reduce the total completion time for a single machine. Furthermore, in this paper, bandwidth rates between users and data centres are analysed for energy-saving and managing a balanced workload using the SSPD-HG approach.

  • Broad comments

Paragraph 1, which is the introduction to the article, should be elaborated more,  concerning definitions of important constructs. Furthermore, not all ‘thoughtsteps’ taken by the authors  are clear in the text (see also my specific comments).

What I miss are definitions of important constructs in the article, like ‘sustainability’, ‘interactive workload’ and ‘batch workload’.

Please look after the English. There are sentences in the text which I do not understand.

  • Specific comments

Line 11: The authors state ‘ Presently, Energy is considering a significant resource that grows scarce with high demand and population in the global market. Therefore, a survey suggested that Renewable energy sources are typically unreliable in accordance with traditional energy sources.’ The second sentence does not logically follow on the first sentence (Why ‘therefore’?).

 Author’s response: It has been incorporated as per the suggestion.

Line 47: The construct ‘sustainable’ hase to be defined and elaborated more.

 Author’s response: The Terms clearly defined “the need to promote the integration of green energy and to reduce energy use costs by balancing regional loads, the timely production of delay-tolerant working loads and the management of thermal storage at data centres”

Line 52: The authors write ‘By comparison to current brown electricity from energy networks, though, green energy is unreliable and uncontrollable from renewable sources, (…)’. Please elaborate why green energy is unreliable and uncontrollable. This is very important, as the authors then come up with a solution fort his problem. But the problem is not clear to me yet.

 Authors response: The main challenge is the task of managing electricity supply and demand instantaneously which can particularly resolve this issue of green energy with large-scale electric energy collected from renewable sources

Line 61: The author write ‘Although it is helpful to manage geographical loads, there are two potential possibilities for promoting the introduction of renewable energies in data centres which is coordinated based on the control center. The difference between an interactive workload and batch workload follows two aspects’. Here the authors go too fast. How is the laste sentence related to the former sentence?

 Authors response: Since it is elaborated in control center it follows as per the potential aspects.

Line 63: Please define what is meant with ‘interactive workload’ and ‘batch workload’.

Authors response: It defines the response time that refers to the output system for the virtual workload;

Line 68: What is meant with ‘stored resources’? You mean energy storage?

Authors response: it meant energy storage as per the context meaning.

Line 95: With ‘IMC algorithm’is meant ‘EMC algorithm’? Otherwise, the authors have to write first in full what is meant by ‘IMC algorithm’.

Authors response: the correlated comment is true, it meant EMC

Line 104: Please write in full first the abbreviation SLA

Authors response: Social level Agreement

Line 118: The authors write ‘The representation of task or job is given by?, m, and n has been modelled’. What does l, m and n stand for? And how do the authors come up with these ‘dimensions’? Where are they for example derived from?

Authors response: it has been formulated in the Eq(5) and Eq(6)

Line 127: Please elaborate more what is meant with ‘the total rest power of sequence of jobs’.

Authors response: completion and total energy consumption time., is the installation and the total marginal energy consumption between jobs m and n.

Line 136: The authors write ‘(…) the information has been taken from Google data centres’. Please explain from the point of view of methodology how the authors did take this information.

 Authors response: It has been taken for data analysis and control

Line 244: What is, in this context, meant with a ‘chromosome’?

Authors response: high energy factor are represented as chromosome’

Round 2

Reviewer 2 Report

The authors have done a lot with my comments. But I still have a problem with paragraph 1, which is the introduction to the article. It should be elaborated more,  concerning the definition of the construct ‘sustainability’, The authors state concerning ‘sustainability’ that ‘The Terms clearly defined “the need to promote the integration of green energy and to reduce energy use costs by balancing regional loads, the timely production of delay-tolerant working loads and the management of thermal storage at data centres”. But this is not my point. From whom is this definition? From the authors? But how is this definition embedded in the literature concerning ‘sustainability’. The authors should have discussed the construct sustainability on the basis of relevant literature, and then come up with a definition which is embedded in this literature.

Also the authors still have to change IMC in line 94 in EMC.

Author Response

Dear Reviewer thanks for your comments. Please find the response to each comment:

The authors have done a lot with my comments. But I still have a problem with paragraph 1, which is the introduction to the article. It should be elaborated more,  concerning the definition of the construct ‘sustainability’, The authors state concerning ‘sustainability’ that ‘The Terms clearly defined “the need to promote the integration of green energy and to reduce energy use costs by balancing regional loads, the timely production of delay-tolerant working loads and the management of thermal storage at data centres”. But this is not my point. From whom is this definition? From the authors? But how is this definition embedded in the literature concerning ‘sustainability’. The authors should have discussed the construct sustainability on the basis of relevant literature, and then come up with a definition which is embedded in this literature.

Ans: The major aim of this research is to reduce the energy emissions of data centers through sustainable energy management at data centers with less delay tolerance. Renewable energies are typically sporadic and volatile, in comparison to conventional electricity energies. Whether to best utilize green electricity in data centers from such clean sources is a problem. In this paper, to promote renewable energy adoption and minimize costs for energy usage by spatial load balancing, opportunistic scheduling of delay-tolerant workloads control in data centers has been accomplished using SSPD-HG.

Further in relation to the comments addressed the related literature has been added form Ref[22,23][27,28,29]

Also the authors still have to change IMC in line 94 in EMC.

Ans: It has been updated